# Scaffold-Free Strategies in Dental Pulp/Dentine Tissue Engineering: Current Status and Implications for Regenerative Biological Processes

**DOI:** 10.3390/bioengineering12020198

**Published:** 2025-02-18

**Authors:** Mohammad Samiei, Martin Conrad Harmsen, Elaheh Dalir Abdolahinia, Jaleh Barar, Xenos Petridis

**Affiliations:** 1Department of Pathology and Medical Biology, University Medical Center Groningen, University of Groningen, P.O. Box 30.001, 9700 RB Groningen, The Netherlands; m.samiei@umcg.nl (M.S.); m.c.harmsen@umcg.nl (M.C.H.); 2Research Center for Pharmaceutical Nanotechnology, Biomedicine Institute, Tabriz University of Medical Sciences, Tabriz 5165665811, Iran; 3Department of Oral Science and Translation Research, College of Dental Medicine, Nova Southeastern University, Fort Lauderdale, FL 33314, USA; elahehdalir@gmail.com; 4Department of Pharmaceutical Sciences, College of Pharmacy, Nova Southeastern University, Fort Lauderdale, FL 33328, USA; jbarar@gmail.com; 5Department of Endodontics, Section of Dental Pathology & Therapeutics, School of Dentistry, National and Kapodistrian University of Athens, 115 27 Athens, Greece; 6Department of Endodontology, Section of Fundamental Dentistry, Center for Dentistry and Oral Hygiene, University Medical Center Groningen, University of Groningen, 9713 GZ Groningen, The Netherlands

**Keywords:** cell spheroids, dental pulp stem cells, regenerative endodontics, scaffold-free tissue engineering

## Abstract

Conventionally, root canal treatment is performed when the dental pulp is severely damaged or lost due to dental trauma or bacterial endodontic infections. This treatment involves removing the compromised or infected pulp tissue, disinfecting the root canal system, and sealing it with inert, non-degradable materials. However, contemporary endodontic treatment has shifted from merely obturating the root canal system with inert materials to guiding endodontic tissue regeneration through biological approaches. The ultimate goal of regenerative endodontics is to restore dental pulp tissue with structural organization and functional characteristics akin to the native pulp, leveraging advancements in tissue engineering and biomaterial sciences. Dental pulp tissue engineering commonly employs scaffold-based strategies, utilizing biomaterials as initial platforms for cell and growth factor delivery, which subsequently act as scaffolds for cell proliferation, differentiation and maturation. However, cells possess an intrinsic capacity for self-organization into spheroids and can generate their own extracellular matrix, eliminating the need for external scaffolds. This self-assembling property presents a promising alternative for scaffold-free dental pulp engineering, addressing limitations associated with biomaterial-based approaches. This review provides a comprehensive overview of cell-based, self-assembling and scaffold-free approaches in dental pulp tissue engineering, highlighting their potential advantages and challenges in advancing regenerative endodontics.

## 1. Background

### 1.1. Dental Pulp: Functions and Importance

The dental pulp is the soft connective tissue of the inner part of the tooth. It is encased in a hard outer shell called dentine. Dental pulp features an extensive neuro-vascular network and connective tissue with fibroblasts, dental pulp stem cells (DPSCs) and specialized post-mitotic odontoblasts [1,2]. Odontoblasts align along the pulpo-dentinal border, and their main function is to deposit dentine throughout the lifespan of a tooth [3,4]. Once a dentine defect occurs and odontoblasts are lost due to dental caries or trauma, quiescent progenitor cells that reside in the perivascular niche of the dental pulp are activated (dental pulp stem cells-DPSCs), migrate to the damaged area, differentiate to hard tissue-depositing cells, and repair the defect [2,3,4,5,6,7]. However, an advancing caries front will eventually surpass the reparative capacity of the dental pulp, inevitably leading to pulp necrosis [5].

Dental pulp necrosis and its sequel, apical periodontitis, are conventionally treated with root canal treatment. Despite the high success of the treatment in preventing and/or resolving apical periodontitis, root canal-treated teeth are deprived of the organ that enables tissue homeostasis, sensory function, and immunocompetency. Additionally, their fracture resistance is decreased as a result of the unavoidable dentine removal occurring during root canal treatment [6]. Epidemiological data suggest that pulpal involvement may hasten tooth loss, especially if root canal treatment is not followed by proper (and often extensive) restoration of the affected teeth [7]. In immature teeth, the impact of pulp necrosis is even more pronounced. Root development ceases when pulp necrosis occurs in immature teeth, increasing the likelihood of catastrophic tooth fractures and tooth loss. The consequences of tooth loss at a young age may be devastating as disturbances in maxillofacial development can lead to esthetic and functional problems that are difficult to treat. Therefore, dental pulp regeneration is particularly warranted for immature teeth [8].

### 1.2. Dental Pulp Regeneration: Clinical Facts and Troubleshooting

Over the past two decades, the concept of regenerative endodontics has grown exponentially [9]. Revascularization of dental pulp by inducing bleeding from the periapical tissues and allowing a blood clot form in the root canals was initially proposed by Nygaard-Ostby [10,11]. These seminal animal experiments laid the groundwork for the clinical translation of revascularization in the early 2000s [12,13]. Soon after the first successful clinical cases of revascularization, the term “regenerative endodontics” replaced revascularization, as it better reflected the biological outcome aspired by the clinical procedure applied [13]. However, actual dental pulp regeneration has proven to be an exception rather than rule with the contemporary regenerative endodontic procedures clinically applied. This has shifted the goals of clinical regenerative endodontics from regeneration to guided hard tissue repair that strengthen the immature root, thus benefiting greatly the young patients [14]. It is now acknowledged that the current regenerative endodontic clinical protocols reproducibly enable the healing of apical periodontitis associated with developing (and fully developed) teeth, while allowing for resumption of the previously ceased root maturation [15,16]. Transitioning from the biologically guided tissue repair to histological and functional reconstitution of the pulp-dentine complex calls for different treatment regenerative and tissue engineering approaches.

In traditional tissue engineering, the base components of tissues are combined, i.e., (stem) cells as building blocks of the living tissue, extracellular matrix (ECM) in the form of biomaterials that support cell attachment and growth, and growth factors that regulate several cell functions. For the effective delivery of cells and/or growth factors into a root canal, various types of scaffolds have been used. Despite the exponential progress in material science and scaffold engineering, multiple design parameters, translatability, and applicability issues (root canal dimensions/niche, chair-side convenience, and overall costs) need to be considered when scaffold-based strategies are applied for in situ dental pulp regeneration [17,18,19].

### 1.3. Scaffold-Free Tissue Engineering: Definitions That Matter

In recent years, tissue engineering without the use of exogenous scaffolds, the so-called “scaffold-free” or “scaffold-less” approach, has been gaining ground as a stem cell-based treatment strategy [20,21]. In dental tissue engineering, scaffold-free methods have several benefits compared to conventional scaffold-based approaches. Scaffold-free methods, such as cell sheets, spheroids, and tissue strands diminish the requirement for fabricating and implanting (bio)materials, thereby curtailing the possibility of immune responses and challenges related to scaffold degradation. Moreover, these approaches enhance natural cell–cell interactions and extracellular matrix synthesis, which are essential for the regeneration of complex dental structures. In dental pulp tissue engineering, scaffold-free approaches have been shown to promote the development of pulp-like tissues with sufficient vascularization and cellular arrangement. Nonetheless, challenges continue to exist particularly in attaining and maintaining the required three-dimensional architecture and mechanical stability essential for functioning dental tissues [22,23]. Scaffold-free approaches require initial interaction between cells to form aggregates, facilitated by their surface adhesion molecules. This initial adhesion triggers various cellular processes, including deposition of a basal membrane, proliferation, differentiation and maturation. During these processes, an interstitial extracellular matrix primarily composed of collagen type I, is also I [24,25]. Contrary to the notion that spheroids are enclosed in their own extracellular matrix, which would impede their ability to sustain continuous proliferation [26], these self-assembled cell aggregates possess the potential to evolve into more extensive, self-organizing structures, similar to the connective tissue framework of the dental pulp. In addition, stem cell-generated spheroids also degrade ECM similar to ECM turnover in physiological tissues. This microstructure supports vessel and neuron ingrowth, fosters cell maturation, and facilitates cell differentiation to odontoblast-like cells and stimulates these to deposit dentine [20,26,27]. In fact, scaffold-free methods are promising for tissue engineering because these partially replicate the developmental processes of tissue formation. In the realm of scaffold-free approaches; however, two distinct biological processes occur namely, self-organization and self-assembly, which warrant further clarification [28,29].

Cellular self-organization refers to the spontaneous formation of organized structures by cells within a growing tissue system through a series of coordinated and overlapping events, including cellular migration, differentiation, and patterning. Cellular response to gradients of signaling molecules, frequently growth factors, is a critical phenomenon in organogenesis, since it directs cell migration and subsequent phenotypic alterations. As cells migrate and differentiate, they also generate signals that engage surrounding cells in proliferation, differentiation, maturation, or death, thereby contributing to the comprehensive development of organs, teeth, and other complicated patterns. The term “self-organization” is often used in the field of dental pulp regeneration to refer to the capacity of dental pulp stem cells to generate functional tissue structures [26,30,31,32]. Unlike epithelial cells, which may spontaneously develop organoids as a result of intrinsic gradients that promote organogenesis (as evidenced in the creation of “mini-guts”) [33], DPSCs, which originate from ecto-mesenchymal cells, do not possess the innate ability to independently arrange themselves into complex structures. Mesenchymal cells, including DPSCs, typically rely on external signals, such as mechanical stimulation or paracrine signals, to initiate further differentiation and maturation. For instance, DPSCs do not undergo spontaneous differentiation into endothelial cells; this developmental process needs exposure to circumstances that promote endothelial growth in in vitro condition [34]. However, a more accurate explanation would acknowledge that this mechanism is mostly dependent on cellular self-assembly, in which cells form cellular aggregates by adhering to each other via cell adhesion molecules (CAMs) such as N-Cadherin and N-CAM. The first aggregation is a crucial stage in tissue development but does not sufficiently include the intricate self-organization seen throughout organogenesis [35,36].

The factors that impact successful cellular self-assembly are the varying cell quantities, usually ranging between 1 million and 100 million cells per mL, depending on the cell type, [37,38] and the final size of the self-assembled tissue construct, which is dictated by diffusion limitations [39]. Of note, the viability of thick (>1 cm) engineered tissues is problematic due to limitations in oxygen and nutrient supply, as well as the removal of toxic waste products, that in most in vitro systems predominantly rely on diffusion [40]. The broadly accepted diffusion limit for cell-dense tissues, such as skeletal muscle and dental pulp is less than 200 μm in vivo [41,42]; hence, rapid vascular ingrowth is essential to re-establish perfusion and ensure survival of the construct in situ [20,43].

### 1.4. Aim

As scaffold-free approaches gain traction in the field of dental pulp regeneration, this review aims to outline the principles of scaffold-free tissue engineering strategies and elaborate on methods for fabricating multicellular building blocks such as cell sheets, cell spheroids, and tissue strands, particularly as applied in the field of dental pulp tissue regeneration/engineering (Figure 1). As well as providing an overview of experimental findings, this review dissects the impact of this scaffold-free strategy on distinct biological events occurring during dental pulp regeneration, such as vascularization, innervation, and dentine formation, in an attempt to shed more light on specific caveats that need to be overcome for successful clinical translation.

## 2. Scaffold-Free Strategies

### 2.1. Cell Sheets

Cell sheet engineering has emerged as a distinctive, scaffold-free technique for tissue regeneration, which can be stimulated by treating cell cultures with ascorbic acid or culturing cells in temperature-responsive cell culture vessels [44,45]. Cell sheets have shown favorable results in the regeneration of various tissues, which is attributed to the lack of proteolytic enzymes such as trypsin and/or dispase during cell culturing. In the absence of these proteolytic enzymes, extracellular matrix (ECM), cell–matrix connections and cell–cell connections are preserved [46,47,48]. Cell sheets have been extensively used in dental tissue regeneration; periodontal ligament cell sheets have effectively facilitated periodontal regeneration, including an acellular cementum-like layer with fibrils anchoring into this layer [49,50]. The use of thermoresponsive substrates is the most attractive and well-studied method for the development of cell sheets [51,52]. In this method, a detachable cell layer that deposits its own ECM is generated on a ~20 nm thick layer of the temperature-responsive poly (N-isopropylacrylamide) polymer (PNIPAm), which is covalently bound to the polystyrene surface of cell culture dishes [52]. Cell sheets, including their own deposited ECM, can then be easily detached from the polymer by lowering culture temperatures without damaging the integrity of the cell sheet or the cell-to-cell and cell-to-ECM adhesions. The underlying mechanism of this technique lies in the hydrophilic behavior of the polymer at different temperatures. Similarly to the surface of commercially available tissue culture dishes, the polymer presents a relatively hydrophobic surface at 37 °C temperature. Below 32 °C, the surface of the polymer becomes hydrophilic, causing the attached cell monolayers to lose affinity and detach from the surface [53]. Using this innovative method, researchers have successfully fabricated cell sheets composed of various cell types, such as cardiac cell sheets [53,54,55,56]. DPSCs-derived sheets have been used for dental pulp regeneration in semi-orthotopic animal models [44,45]. Cylindrically shaped DPSCs sheets were inserted into empty, human-derived root canal segments and subsequently implanted in immunocompromised Balb/C nude mice. The newly formed root canal tissues showed vascularization and expression of dentin sialoprotein in odontoblast-like cells aligned along the root canal wall [57]. In a similar model, cell sheet-derived pellets (CSDP) fabricated from stem cells from the apical papilla (SCAPs) formed tissues resembling the dental pulp/dentine complex in human-derived root canals [58].

### 2.2. Cell Spheroids

In tissue engineering and regenerative medicine, spheroids are used as implantable therapeutic agents, often in preclinical studies involving animal models. Cells within the spheroids demonstrate improved cell viability, protein secretion, and the potential for stem cell differentiation. These advantages are attributed to the three-dimensional spherical shape of the spheroids, which enables extensive chemical and mechanical interactions between the cells and the extracellular matrix [59,60]. Particularly in the field of dental pulp regeneration/engineering, cell spheroids have been used as regenerative building blocks in research on various animal models [61]. Spheroids are generated by the aggregation of individual cells in biological fluids, facilitated by cell adhesion molecules, extracellular matrix proteins, and integrins [62]. The development of multicellular spheroid (MCS) consists of three essential stages [63]: (i) extracellular matrix (ECM) fibers with receptor-gated dimer (RGD) patterns attach to integrins on the cell surface, cells form loose clusters, which results in elevated production of cadherin, (ii) cadherin sequesters at the cell membrane cell adhesion is increased, (iii) cells undergo compaction into solid aggregates by the process of cadherin-cadherin binding [64]. Upon maturation, spheroids usually form a tripartite structure consisting of an outer proliferative rim, a middle quiescent layer, and a necrotic core [62]. A proliferative rim consists of cells undergoing active division, whereas a necrotic core develops as a result of constraints in nutrition and oxygen transport [65]. In contrast to organoids, spheroids produced from mesenchymal stem cells are structurally layered and lack a lumen. Their development primarily occurs in the outer layers [66]. Currently, various techniques are employed for creating cell spheroids. These include methods like pellet culture, spinner culture, hanging drop (HD), liquid overlay, rotating wall vessel, external force, microfluidics, micro molded non-adhesive hydrogels [67], microwell culture, medium regulation, bioreactors [68], and bioactive materials like cellulose hydrogel film [69]. Regardless of the technique employed, the resulting DPSC spheroids typically exhibit a spherical or spheroid-like morphology with variable diameters. The viable outer layer of cells in these spheroids usually measures around 100–300 μm [70]. It is worth mentioning that spheroid size is a major determinant of its biological behavior. For instance, HepG2 spheroids measuring 200 μm in diameter exhibited the highest albumin secretion compared to those with a diameter of 300 μm or larger [71]. Three-dimensional cultivation of DPSC spheroids results in increased expression of stemness and pluripotency markers (Oct4, Sox2, NANOG, TP63) compared to conventional 2D cultures. Immunofluorescence staining and histological analysis has revealed that these markers are primarily localized in the central and intermediate zones of the spheroids, suggesting that certain areas within the spheroids contain cells with elevated marker expression [72,73]. Additionally, DPSC spheroids differentiate into functional neuron-like cells under neurogenic induction conditions in vitro [74]. Notably, the expression of neuronal markers, such as microtubule-associated protein 2 (MAP2), increased in DPSC spheroids following 3D culture in neurogenic medium or with the addition of epidermal growth factor (EGF) and basic fibroblast growth factor (bFGF) [75,76,77]. In one study, pre-vascularized spheroids of 300 μm in diameter, consisting of co-cultured DPSCs and human umbilical endothelial cells (HUVECs) were fabricated in agarose mold microwells [48]. In vitro experiments showed that DPSCs promoted the survival of co-cultured human umbilical endothelial cells in the microtissue spheroids. When these pre-vascularized, microtissue spheroids were inserted into the root canal space of a root segment and implanted subcutaneously in immunodeficient mice, a highly vascularized, cellular tissue resembling dental pulp was formed within the root segment. Histological and immunohistochemical examinations verified the human origin of the regenerated tissue. Notably, odontoblast-like cells were arranged along the dentine, while the vascular structures were effectively incorporated into the host vasculature [61].

### 2.3. Tissue Strands

Tissue strands are one of the most recent developments in scaffold-free building elements for tissue engineering. The fabrication of these strands involves administering cell suspensions into a tubular alginate capsule, which is then dissolved using a sodium citrate solution, resulting in the retention of the cell-based tissue strand (Figure 2). This process allows for the linear arrangement of cells within the tissue strands [78]. Tubular capsules function as storage units for cell aggregation, resulting in the formation of extended, flawlessly shaped cylindrical mini-tissues once the capsules are dissolved. These tissue strands are subsequently modified and can be effectively incorporated into a distinctive printing system, facilitating the bioprinting process and swift creation of tissues suitable for applications in regenerative medicine, drug screening, and disease modeling [79]. Capsule materials are essential in tissue strand manufacturing as they serve as a temporary mold that facilitates cell aggregation and maturation prior to removal. Alginate-based capsules are often used in this method owing to their biocompatibility and simplicity of extraction. Nonetheless, these components are not included into the final construction, since tissue strands are intended to be completely scaffold-free [80]. The integration of many cell types into a single strand enables the creation of intricate tissue structures. Unlike conventional bioinks, the tissue strands are composed only of cells, making them appropriate for bioprinting applications that involve the direct deposition of cells without the need for supplementary material. The use of computer-aided design (CAD) technology enables the predefined arrangement of individual tissue strands inside complex three-dimensional (3D) structures, hence increasing the possibility of developing biomimetic tissues. The rapid self-assembly of tissue strands facilitates cellular union and maturation. These distinctive capabilities make it possible to scale up tissue constructs, and they have the potential to integrate vascular networks, paving the way for the bottom-up fabrication of clinically relevant-sized organs solution [81]. Although attaining desirable mechanical characteristics in scaffold-free structures remains a challenge [80,81], tissue strands have shown the most potential for quickly producing biomimetic tissues [82,83].

Recently, Itoh and colleagues conducted a study to investigate the feasibility of utilizing scaffold-free rod-shaped cell construct consisting of DPSCs for pulp regeneration therapy. The rod-shaped design is more suited for efficient pulp regeneration due to its ability to fit readily into the tight and elongated structure of the tooth cavity. They developed a technique to create rod-shaped structures with embedded cells using a thermoresponsive hydrogel (sheet-like aggregates) and examined their capacity for growth, proliferation and odontoblastic differentiation in vitro. This approach allowed for the fabrication of DPSC constructs in various sizes and shapes based on computed tomographic image design [84]. The same research group investigated the potential of DPSCs to contribute to vascular structure and enhance dental pulp regeneration in pulpless teeth. The study aimed to utilize the endothelial differentiation capability of DPSCs to create vascularized DPSC constructs. Endothelial differentiation in DPSCs was induced and their behavior, internal structure, and pulp regenerative capacity was examined in vivo [85]. The results revealed that DPSCs expressing endothelial markers formed a lumen structure at the outer layer of constructs, indicating successful differentiation into endothelial cells and the formation of vascular-like structures. The vascularized DPSC constructs were implanted into human pulpless teeth and subsequently implanted in the dorsal subcutaneous space of immunodeficient mice. After six weeks, the implanted pre-vascularized DPSCs constructions exhibited a notably greater density of human CD31-positive blood vessels in comparison to the non-pre-vascularized constructs. Crucially, both the constructions that were pre-vascularized and those that were not pre-vascularized exhibited the development of blood vessels. Nevertheless, the absence of substantial color changes in the control group and the absence of cavities shown in micro-CT images suggested that the vessels in the non-pre-vascularized group were fewer and most likely derived from the host (mouse) vasculature. The pre-vascularized DPSCs constructions, which included cells covered with endothelial cells, formed a larger network of blood vessels that effectively linked with the host’s systemic circulation. By including both human and mouse endothelial cells, this hybrid vascularization greatly improved the regeneration of pulp-like tissue [85].

### 2.4. Evaluation of Scaffold-Free Stratergies: Technical Features, Biological Characteristics, Regenerative Potential and Clinical Applicability

Scaffold-free methods yield cellular structures that have distinctive features regarding cellular organization and ECM synthesis, which impact differently on their regenerative potential. Cell sheets, cell spheroids, and tissue strands vary in form (shape and size), manufacturing processes, and regenerative capacity. The selection of a suitable scaffold-free technique for dental pulp regeneration must take into account parameters such as biological efficacy and clinical applicability. Cell spheroids and tissue strands exhibit enhanced vascularization and neurogenesis potential, principally attributable to their three-dimensional architecture, which promotes increased cell signaling as well as integration with host tissues. Conversely, cell sheets provide a straightforward and therapeutically applicable method owing to their maintenance of cell–cell connections and extracellular matrix, facilitating their use in regenerative dentistry without necessitating intricate biofabrication processes. Table 1 provides a summary of the basic features, biological characteristics, regenerative potential and clinical applicability of the scaffold-free methods.

## 3. Impact of Scaffold-Free Approaches on Specific Regenerative Biological Processes

Vascularization, neural network development, and dentine deposition are three distinct biological processes that need to take place in a well-coordinated spatiotemporal manner for genuine dental pulp regeneration to occur. Contemporary scaffold-free dental pulp tissue engineering approaches rely on exogenously manipulated dental stem cells (Figure 3). These cells, under specific conditions and the influence of appropriate cues, can fully or partially support the biological processes necessary for reconstituting the functions of the dental pulp. An overview of the dental-derived stem cell populations that can contribute to each of the biological processes mentioned above, irrespective of their use in dental pulp tissue engineering approaches, is presented in Table 2.

### 3.1. Scaffold-Free Strategies and Vascularization in Dental Pulp Regeneration

For successful dental pulp regeneration, a supportive microenvironment must be established within the confined pulp cavity to enable rapid vascularization [101]. The formation of a capillary network is crucial for the survival of implanted constructs, as it facilitates the exchange of oxygen, nutrients, and waste products. Hypoxia and growth factors, such as vascular endothelial growth factor-A (VEGF-A) and angiopoietin-2 (Ang-2), play pivotal roles in triggering this vascularization process [102]. Mechanistic studies utilizing scaffold-free techniques have highlighted the potential of dental pulp stem cells to enhance vascularization. For example, DPSCs organized in spheroids or sheet-based cultures, rather than embedded in scaffolds, have shown improved capacity for vascular network formation when co-cultured with endothelial cells such as human umbilical vein endothelial cells (HUVECs) [93,103]. These scaffold-free systems promote cell–cell interactions and preserve the native signaling mechanisms essential for vascularization. Notably, scaffold-free co-cultures of DPSCs and HUVECs result in better vascular outcomes compared to respective mono-cultures [94,104]. A key study demonstrated that TGF-β1-treated DPSCs in a spheroid model stabilized newly formed blood vessels via Ang1/Tie2 and VEGF/VEGFR2 signaling pathways [93]. TGF-β1 stimulated the phosphorylation of Smad2 and Smad3, upregulating Ang1 and VEGF production. Secreted Ang1 subsequently activated Tie2 receptors on HUVECs, enhancing VE-cadherin expression and promoting endothelial cell adhesion, which stabilized the vascular structures. In these models, DPSCs further differentiated into pericyte-like cells, aiding vessel stability and integration with host vasculature [93]. One study investigated the roles of DPSCs derived from scaffold-free conditions: E-DPSCs, obtained from direct co-culture with HUVECs, and T-DPSCs, treated with TGF-β1, in stabilizing blood vessels both in vitro and in vivo [94]. In a 3D spheroid sprouting assay, both E-DPSCs and T-DPSCs exhibited smooth muscle cell-like properties, expressing higher levels of mural cell-specific markers and suppressing HUVEC sprouting. Mechanistically, they inhibited HUVEC sprouting by activating the Ang1/Tie2/VE-cadherin signaling pathway and down-regulating VEGFR2 expression, thereby promoting vessel stability [94]. Moreover, emerging studies using microfluidic “pulp-on-chip” platforms have shown that scaffold-free co-cultures of stem cells from apical papilla (SCAPs) and HUVECs can recreate the spatial organization of vascular and neural networks. These systems demonstrated efficient vessel-like structure formation and neural invasion into vascular clusters, highlighting the potential of scaffold-free models for studying vascularization and its interplay with other regenerative processes [105]. These findings underscore the significance of scaffold-free techniques in understanding and enhancing vascularization mechanisms. By overcoming the limitations of artificial scaffolds, these approaches maintain the native signaling and interactions critical for the stabilization and integration of vascular networks during dental pulp regeneration.

### 3.2. Scaffold-Free Strategies and Neurogenesis in Dental Pulp Regeneration

Considering the neural crest origin of the dental pulp [106], it is not surprising that dental pulp-derived stem cells express mature neuronal markers (β III-tubulin and NeuN) and neural stem cell markers (Nestin) at basal conditions and after neurogenic induction [107,108]. Compared to 2D cultures, formation of scaffold-free 3D neurospheres enhances the neurogenic potential of stem cells from apical papilla (SCAPs), as confirmed by morphological changes, marker expression and functional characteristics [109,110]. In light of this evidence, scaffold-free protocols aiming at optimizing DPSCs neurosphere formation and maximizing their neurogenic differentiation capacity have been investigated [111]. Nevertheless, clinical application of dental pulp-derived neurospheres in dental pulp tissue regeneration/engineering is yet to be investigated, which creates exciting opportunities for future research.

On the other hand, the clinical application of cell sheet engineering seems a more tangible goal. Cell sheets derived from dental pulp cells secrete neurotrophic factors which mediate an effective axon regeneration [112,113]. This neurotrophic effect may account for the reinnervation demonstrated in regenerated dental pulp tissues removed from the root canals of teeth from animal models and humans [114,115]. Nonetheless, the underlying mechanisms behind this cell sheet-boosted neurotrophic behavior of dental pulp cells remain obscure, which warrants further mechanistic investigations. In a clinical scenario, scaffold-free, dental pulp-derived cell sheets could act as the delivery vehicle of neurotrophins that would facilitate the ingrowth of neural axons into the root canals from the innervated periapical region resulting and the establishment of nerve conduits in the regenerated dental pulp.

### 3.3. Scaffold-Free Strategies and Dentine Formation in Dental Pulp Regeneration

A successfully regenerated dental pulp should contain cells morphologically and functionally similar to odontoblasts, with spatial distribution at the dental pulp periphery, in order to serve its dentine secretory function. The biochemical pathways involved in the differentiation of DPSCs into functional odontoblasts are similar to differentiation pathways of bone marrow mesenchymal stem cells (BMSC) into osteoblasts [116].

Increased expression of dentinogenic-related genes such as OCN, DSSP, and ALP genes has been demonstrated in DPSCs-derived spheroids when compared to 2D DPSCs cultures [117]. In a semi-orthotopic mouse model, cell sheet-derived pellets comprising human SCAPs have regenerated a pulpo-dentinal complex as evidenced by the deposition of new dentine on root dentine fragments; new dentine was laid down from newly differentiated odontoblast-like cells of human origin residing in a well-vascularized, neo-formed dental pulp-like tissue [58]. In addition, findings from experimental studies on tooth root engineering have demonstrated the dentinogenic potential of cells sheets comprising SHED, DPSCs and dental follicle cells (DFCs) under the modulating influence of the bioactive and mechanical cues provided by a demineralized, stiff dentine matrix [90,118,119]. Taken together, dentinogenesis seems a tangible goal when properly instructed scaffold-free, dental stem cell-derived 3D cell cultures are utilized. From another perspective, spheroids may induce dentinogenesis rather than generating new dentin directly. Hertwig’s epithelial root sheath (HERS) cells drive dentinogenesis during tooth development via their interaction with the ecto-mesenchymal cells of the dental pulp [27,120]. When compared to 2D cultured HERS, HERS spheroids induced a more potent dentinogenic differentiation of dental pulp cells. The ectopic transplantation of co-cultured HERS spheroids and dental pulp cells in a rodent model resulted in the formation of highly ordered mineralized tissue resembling dentin, in contrast to the disorganized mineralization observed with 2D HERS cell co-cultures. The hypoxia-inducible factor-1 (HIF-1) pathway, which plays a critical role in HERS spheroid formation and function, could further support their dentinogenic potential [121].

## 4. Infection Control Strategies in Scaffold-Free Regenerative Endodontics

To ensure the efficacy of scaffold-free regenerative endodontic treatment, the previously infected intraradicular space must be adequately disinfected, as microbial persistence is the leading cause of treatment failure in regenerative endodontics [122,123]. The fact that scaffold-free regenerative endodontics utilizes exogenous cells and does not rely on the recruitment of mesenchymal stromal cells residing in the apical papilla and/or the periapical tissues (currently clinically practiced regenerative endodontic procedures) may allow clinicians to leverage the maximum disinfection potential of potent disinfectants, merely by increasing their concentration (e.g., >5.25% sodium hypochlorite compared to the low concentrations currently recommended); this disinfection regime is not currently recommended due to the cytotoxic effects of higher concentrations of sodium hypochlorite to endogenous apical papilla stem cells [124]. Furthermore, antimicrobial photodynamic therapy (aPDT), which employs light-activated photosensitizers to generate reactive oxygen species for bacterial eradication may provide an effective supplementary method to enhance root canal disinfection prior to implanting scaffold-free cellular structures [125]. Lastly, the development on next generation antibacterial strategies, mainly in the form of novel nanoparticle- or hydrogel-based formulations intended for intraradicular placement, could optimize root canal disinfection [126].

## 5. Current Limitations of Scaffold-Free Approaches

Although scaffold-free approaches in dental tissue engineering offer distinct benefits, each method has challenges that must be resolved for effective clinical use. Cell sheets, while maintaining cell–cell connections and extracellular matrix, lack intrinsic three-dimensional design, hence restricting their mechanical stability and vascularization potential, which may result in hypoxia-induced cell death in thicker structures [127]. Moreover, their delicate nature makes accurate transplantation complicated. Cell spheroids provide improved cell–cell interactions and replicate in vivo microenvironments; yet, they have size-dependent viability challenges, as bigger spheroids encounter restricted oxygen and nutrient transport, resulting in necrotic core development [61]. Furthermore, attaining consistent spheroid dimensions presents technical difficulties, impacting repeatability in regenerative applications, while their absence of structural order may impede integration within the dentin-pulp complex [118]. Tissue strands, as a novel scaffold-free approach, demonstrate mechanical instability owing to insufficient cellular unity, which affects construct integrity following implantation. The development of functional vascular networks is a significant difficulty, since these structures need fast vascular integration for in vivo survival, and scaling presents difficulties in maintaining uniform cell density and alignment along the strand [80,128]. Although scaffold-free methods show considerable promise, it is crucial to address these issues by enhancing vascularization procedures, improving mechanical properties, and developing scalable biofabrication processes to promote their clinical application.

## 6. Future Perspectives

Scaffold-free approaches such as cell sheets, spheroids, and more recently, tissue strands are under scrutiny for their impact on dental pulp tissue regeneration/engineering. Fabricating pre-vascularized, scaffold-free tissue constructs by combining stem cells with endothelial cells or by guiding single dental stem cell populations with appropriate cues is feasible. The scaffold-free, newly formed microstructures secrete natural ECM and develop stable vascular networks that can rapidly connect with the host vasculature. Dental stem cell-based scaffold-free formations can effectively support the distinct biological processes associated with in situ dental pulp regeneration. Nonetheless, and despite the promising results coming from in vitro investigations, ectopic, and semi-orthotopic animal models, there is still no solid scientific evidence for the use of scaffold-free strategies in the clinical setting. Fine-tuning technical parameters (e.g., initial cell density, cell co-cultures, spheroid size) and gaining more insights into the biological mechanisms underlying the behavior of the self-assembled cells within the scaffold-free constructs (e.g., hypoxic conditions/gradients developed within the spheroids, amount and type of ECM produced) will help us exploit the advantages of scaffold-free tissue engineering to their maximum along with successfully translating the in vitro and preclinical findings into the clinical settings.

Currently, there is no standardized, reliable protocol for the in vivo application of one scaffold-free method over another in translational dentistry. Future investigations may show that a hybrid engineered construct comprising more than one of the scaffold-free approaches highlighted in this paper integrates all the biological and mechanical characteristics needed for dental pulp regeneration. Lastly, consolidating scaffold-free with scaffold-based strategies could integrate the advantages of both. In the future, in situ bioprinting of a biomimetic decellularized dental pulp matrix hydrogel scaffold consisting of pre-vascularized SHED spheroids, DPSCs neurospheres and dentinogenic cells sheets may address all the demands of true dental pulp regeneration.

## Figures and Tables

**Figure 1 bioengineering-12-00198-f001:**
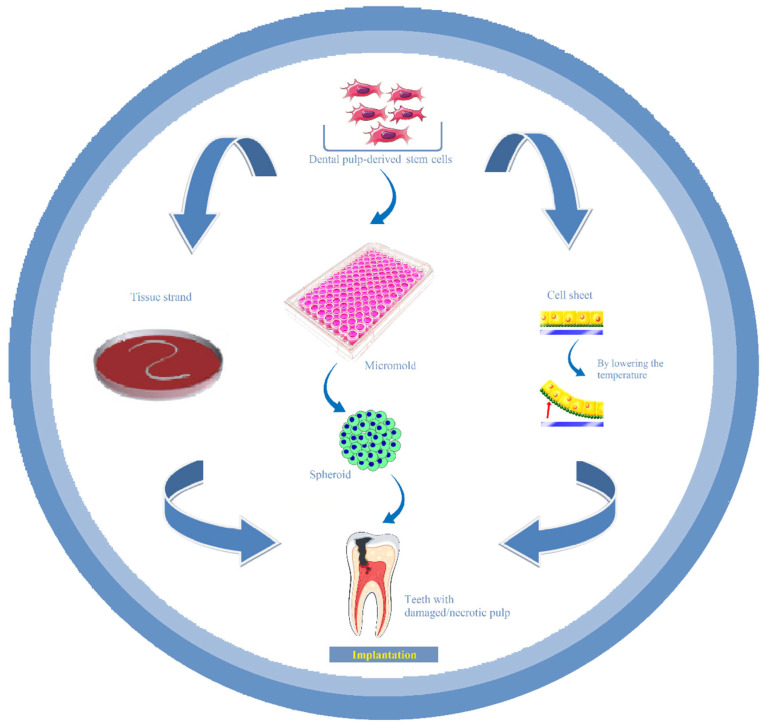
Overview of scaffold-free strategies used for in situ dental pulp regeneration. Cell sheets, spheroids and tissue strands, with dental pulp-derived stem cells as building blocks, can be implanted in the root canals of pulpless teeth to regenerate the missing dental pulp tissue.

**Figure 2 bioengineering-12-00198-f002:**
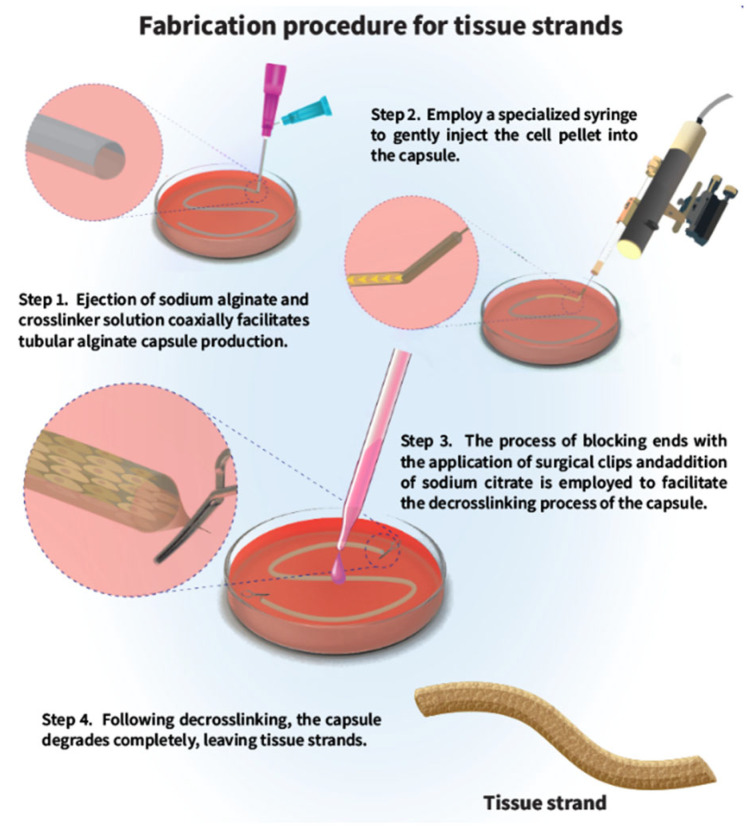
Tissue strand fabrication steps. In the first step, sodium alginate is ejected coaxially, and the crosslinker solution facilitates the fabrication of tubular alginate capsules. The second step involves gently aspirating the cell pellet into the capsule using a custom syringe to minimize damage to the cells during transfer. The third step involves culturing the cell pellet for a few days after blocking ends with surgery clips to facilitate cell aggregation and following that adding sodium citrate to decrosslinking the capsule. Finally, the capsule degrades completely, leaving tissue strands behind.

**Figure 3 bioengineering-12-00198-f003:**
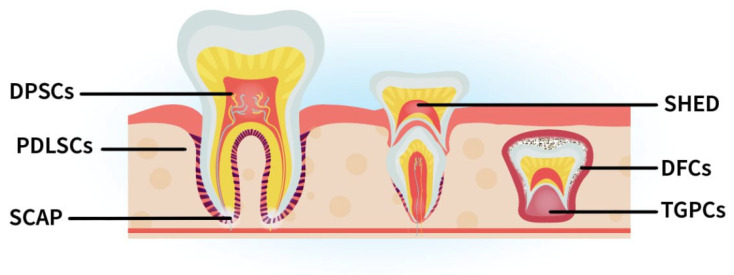
Dental-derived stem cells and their origin. Stem cells from: dental pulp (DPSCs); dental follicle (DFCs); human exfoliated deciduous teeth (SHED); periodontal ligament (PDLSCs); apical papilla (SCAPs); tooth germ-Tooth Germ Progenitor Cells (TGPCs).

**Table 1 bioengineering-12-00198-t001:** Comparison of scaffold-free strategies in dental tissue engineering based on key characteristics and regenerative potential.

Feature/ Characteristic	Cell Sheets	Cell Spheroids	Tissue Strands
Technique	Uses temperature-responsive polymer surfaces to detach cell layers without enzymes.	Cells aggregate on non-adhesive surfaces or bioreactors to form compact spheroids.	Cells encapsulated in alginate tubes, which are later dissolved to form strands.
Shape and Size	Flat, multilayered sheets with micrometer thickness and centimeter-scale dimensions.	Spherical structures, typically 100–300 μm in diameter.	Cylindrical strands with millimeter-scale diameters and variable lengths.
Cell–Cell Interaction	High, due to layered stacking of cells.	High, compact 3D cell aggregates allow stronger interactions.	Moderate, linear arrangement limits direct interactions.
Extracellular Matrix (ECM) Production	Moderate, dependent on cell type and culture conditions.	High, as cell aggregation enhances ECM deposition.	Moderate to high, depending on cell type and culture conditions.
Vascularization Potential	Moderate, can be improved with growth factors.	High, due to enhanced paracrine signaling and microvascular structure formation.	High supports vessel integration into engineered tissue.
Neurogenesis Potential	Moderate, can be enhanced with neurotrophic factors.	High, supports differentiation into neuron-like cells.	High, facilitates neuronal ingrowth and connectivity.
Dentin regeneration potential	Moderate, with variable outcomes across studies.	High, induces odontoblast-like differentiation and dentin matrix formation.	High, supports dentin matrix deposition and mineralization.
Ease of Clinical Translation	High, simple to prepare and transplant.	Moderate, requires optimization for stability and integration.	Lower, due to handling complexity.

**Table 2 bioengineering-12-00198-t002:** An overview of scaffold-free strategies employing dental stem cell populations (DSCs) for promoting vascularization, neurogenesis and mineralization. References with an asterisk (*) indicate in vivo (animal model) studies exploring the use of scaffold-free strategies for dental pulp regeneration.

Scaffold-Free Method	Cells	Experimental Model	Regenerative Process (Angiogenesis, Neurogenesis, Mineralization)	Basic Findings	References
Cell sheet	Human DFCs	rat	mineralization	Implanting Allogeneic TDMP/DFCs sheets cell sheets in TDM ^1^-containing media increased bone development (osteogenesis) in rat and canine calvarial bone deficiency models.	[86]
Cell sheet	Human DFCs	immunodeficient mice	mineralization	Osteogenic differentiation capacity of DFCs sheets is superior to PDLSCs sheets.	[87]
Cell sheet	Human SCAPs	immunodeficient mice	mineralization	Subcutaneous SCAP cell sheets were implanted in immunocompromised mice. A well-vascularized, dentine-coated tissue developed after 6 weeks.	[58] *
Cell sheet	Human PDLSCs	in vitro	mineralization	The impact of rutin on PDLSC sheet development and osteogenic differentiation was examined. PDLSC cell sheets treated with rutin were denser/richer in ECM, and more osteogenic.	[88]
Cell sheet	Human PDLSCs	immunodeficient mice	mineralization	USCs ^2^ effects on PDLSC proliferation and differentiation have been examined. The ratio of 1 PDLSC: 2 USC produced denser collagen layers without cell sheet contact and improved osteogenic and cementogenic protein expression.	[89]
Cell sheet	SHEDs	immunodeficient mice	mineralization	DFC and SHED cell sheets expressed osteogenic and odontogenic markers.	[90]
Cell sheet	human DPSCs	in vitro	mineralization	The cells within the hDPSCs cell sheets demonstrated viability for a minimum of 96 h. Furthermore, these cells maintained their stem cell characteristics and preserved their ability to undergo osteogenic differentiation.	[91]
Spheroid	Human DPSCs	immunodeficient mice	angiogenesis	After implanting HUVECs ^3^ and DPSC-containing spheroids in immunodeficient animals’ root canals, a totally vascularized dental pulp-like tissue with odontoblast-like cells emerged after 4 weeks.	[61] *
Spheroid	human DPSCs	in vitro	neurogenesis	DPSCs generated neurospheres in EGF, bFGF, and heparin medium. EGF and bFGF increased Nestin and SOX2 ^4^ expression within 72 h.	[76]
Spheroid	human DPSCs	in vitro	neurogenesis	The optimal cell culture conditions for DPSC cell growth and differentiation in neurospheres were examined. In xeno-free conditions ^5^, DPSC differentiated into neuron-like cells much more successfully.	[92]
Spheroid	Human DPSCs	in vitro	angiogenesis	The application of transforming growth factor beta 1 (TGF-β1) to dental pulp stem cells (DPSCs) resulted in an augmentation of pericyte-like characteristics. These TGF-β1-treated DPSCs (T-DPSCs) demonstrated the ability to stabilize recently formed blood vessels in a 3D co-culture alongside HUVEC cells.	[93]
Spheroid	human DPSCs	SCID mouse model	angiogenesis	The outcomes of a 3D coculture spheroid sprouting and in vivo assays indicated that both E-DPSCs and T-DPSCs demonstrated their capacity to stabilize recently formed blood vessels and improve perfusion.	[94]
Spheroid	human DPSCs	in vitro	mineralization	Berberine’s influence on DPSC spheroids’ osteogenic differentiation was examined. Gerberine dose-dependently enhanced osteogenic marker expression in dexamethasone-induced DPSCs spheroids through the EGFR-MAPK-Runx2 signaling pathway.	[95]
Spheroid	SHEDs	in vitro	neurogenesis	DFC and SHED were cultured in four SRM ^6^ mediums. In the same settings, SHED and DFC had varying neurogenic differentiation capacities and decreased cell proliferation.	[96]
Spheroid	SHEDs	in vitro	neurogenesis	Rho-kinase inhibitors (Y-27632, Noggin) were tested on SHED proliferation and neurogenic differentiation in neurospheres. Cell proliferation, survival, and neurosphere formation enhanced with either inhibitor or mixture.	[97]
Spheroid	Human PDLSCs	immunodeficient rat	angiogenesis	Spheroid 3D culture on PDLSC has been examined. Three-dimensional culture expressed more anti-inflammatory and angio-genesis genes than 2D culture.	[98]
Spheroid	Human PDLSCs	in vitro	neurogenesis	PDLSC development into neurons was examined in 3D culture. Nerve and retinal neurons differentiated from PDLSC-derived neurospheres.	[99]
Spheroid	Human SCAPs	in vitro	- angiogenesis mineralization	SCAP spheroids treated with LBL-cAMP ^7^ showed VEGF production for 14 days and increased gene expression of osteogenic genes.	[100]
Tissue strands	Human DPSCs	immunodeficient mice	- dental pulp regeneration - angiogenesis mineralization	After implantation, the human root canal showed the development of pulp-like tissues containing numerous blood vessels. Also, DPSCs differentiated into odontoblast-like mineralizing cells. Furthermore, the regenerated tissue featured a concentration of human CD31–positive endothelial cells at its core.	[84] *
Tissue strands	vascularized DPSCs	immunodeficient mice	- dental pulp regeneration angiogenesis	DPSCs located in the outer layer underwent differentiation into endothelial cells, giving rise to vascular-like structures within the cellular framework. The development of a vascular structure in the DPSC construct played a crucial role in supporting blood supply and promoting improved pulp regeneration.	[85] *

^1^ Treated dentin matrix; ^2^ Human urine-derived stem cells; ^3^ Human umbilical vein endothelial cells; ^4^ SRY (sex deciding area Y)-box 2; ^5^ Xeno-free conditions indicate presence only of human-derived components and absence of any animal-derived proteins in culture media; ^6^ Serum-replacement media; ^7^ Layer-by-layer self-assembly with gelatin and alginate polyelectrolytes.

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
