# Peer review of "Scaffold-Free Strategies in Dental Pulp/Dentine Tissue Engineering: Current Status and Implications for Regenerative Biological Processes"

_bioengineering, 2025, doi:10.3390/bioengineering12020198_

Round 1
Reviewer 1 Report
Comments and Suggestions for Authors
The review provides comprehensive insights in scaffold-free approach for pulp regeneration, which is helpful for both scientific and translational research. However, there are some issues that should be supplemented:
Please add some description about why scaffold-free approach is more desirable than tissue engineering with scaffolds. Or, what’s the disadvantages and limitations of using scaffold approach.
Please add a table listing and comparing characteristics of cell sheet, cell Spheroid and tissue strands. eg: techniques, shape or size, effectiveness in regenerating dentine, et al.
Please add discussion about the current limitations for each approach, cell sheet, cell Spheroid and tissue strands.
Author Response
Dear Referee,
We sincerely appreciate the time and effort invested in evaluating our work. Your insightful comments and constructive feedback have greatly helped us improve the quality and clarity of our work.
Below, you will find our point-by-point response to each of your comments.
Comment 1:
Please add some description about why scaffold-free approach is more desirable than tissue engineering with scaffolds. Or, what’s the disadvantages and limitations of using scaffold approach.
Authors' Response:
Thank you for pointing this out. In paragraph "1.3. Scaffold-free tissue engineering: definitions that matter", pp. 2-3, lines 98-109, a paragraph, highlighted in yellow, has been added, to address your suggestion.
Comment 2:
Please add a table listing and comparing characteristics of cell sheet, cell Spheroid and tissue strands. eg: techniques, shape or size, effectiveness in regenerating dentine, et al.
Authors' response:
pp. 8-9: Table 1 (highlighted in yellow) has been added as per referees' suggestion.
Comment 3:
Please add discussion about the current limitations for each approach, cell sheet, cell Spheroid and tissue strands.
Authors' response:
pp.14, lines 512-532: a paragraph, titled "Current limitations of scaffold-free approaches" has been added to address the referees' point.
Reviewer 2 Report
Comments and Suggestions for Authors
Figure 2 can be improved to clearly show the capsule and cells in the capsule. There is no need to show the whole culture plate.
In Table 2, the repeated full names of cell types are not necessarily shown since the abbreviations have been indexed in the previous text and Figure 3 caption. Footnote of "7Three-dimensional; 8Two-dimensional" are not necessary.
The review of biomaterials supporting the cell-free process should be reviewed, including the material supporting cell sheet, capsule material composition, properties, and development trend.
Line 159 “on a ~20 nm thick layer ”, 20 nm is a very thin layer and may not form a continuous layer. Please double check the source of the information.
Bacterial infection is a key issue for endodontic treatment failure. Author should review and discuss the approach to prevent endodontic infection in the scaffold-free regenerative endodontics concept.
Author Response
Dear Referee,
We sincerely appreciate the time and effort invested in evaluating our work. Your insightful comments and constructive feedback have greatly helped us improve the quality and clarity of our work.
Below, you will find our point-by-point response to each of your comments.
Comment 1:
Figure 2 can be improved to clearly show the capsule and cells in the capsule. There is no need to show the whole culture plate.
Authors' response:
Figure 2 has been modified as per referee's suggestions.
Comment 2:
In Table 2, the repeated full names of cell types are not necessarily shown since the abbreviations have been indexed in the previous text and Figure 3 caption. Footnote of "7Three-dimensional; 8Two-dimensional" are not necessary.
Authors' response:
Table 2 has been modified as per referee's suggestions.
Comment 3:
The review of biomaterials supporting the cell-free process should be reviewed, including the material supporting cell sheet, capsule material composition, properties, and development trend.
Authors' response:
pp.6, lines 271-275: text has been added (highlighted in yellow) to address the referee's comment.
Comment 4:
Line 159 “on a ~20 nm thick layer ”, 20 nm is a very thin layer and may not form a continuous layer. Please double check the source of the information.
Authors' response:
The source has been checked and provided as attachment below (please see the attachment). The pertinent information has been highlighted in yellow within the pdf (page 43, second column).
Comment 5:
Bacterial infection is a key issue for endodontic treatment failure. Author should review and discuss the approach to prevent endodontic infection in the scaffold-free regenerative endodontics concept.
Authors' response:
pp. 13-14: a paragraph titled "4. Infection control strategies in scaffold-free regenerative endodontics" has been added to address the referee's comment.
